# Sense of Control and Depression during Public Health Restrictions and the COVID-19 Pandemic

**DOI:** 10.3390/ijerph192114429

**Published:** 2022-11-03

**Authors:** Rachel Msetfi, Diana Kornbrot, Yemaya J. Halbrook, Salha Senan

**Affiliations:** 1Department of Psychology, University of Limerick, V94 T9PX Limerick, Ireland; 2Department of Psychology, University of Hertfordshire, Hatfield AL10 9EU, UK; 3Department of Physical Education and Sports Sciences, University of Limerick, V94 T9PX Limerick, Ireland; 4Department of Psychology, King Abdulaziz University, Jeddah 22254, Saudi Arabia

**Keywords:** depression, sense of control, COVID, pandemic, public health restrictions

## Abstract

Depression rates have increased significantly since the onset of the COVID-19 pandemic, with a number of factors implicated in this increase, including stress, fear, social isolation and the psychological impact of public health restrictions. The main purpose of the current cross-sectional survey study was to examine the relationship between the experience of public health restrictions, the sense of control and depression, both during and after restrictions were lifted. A survey methodology was chosen, with data collected in the Republic of Ireland at two time points (January 2022 and May 2022). Time 1 participants (*n* = 314) were invited to repeat the measures 5 months later, with 172 agreeing to be recontacted, and 47 participants completing all measures at two time points. Findings showed that both the sense of control, in relation to perceived constraints, *w* = 0.43, and the experience of restrictions, *w* = 0.14, predicted depression at Time 1. Participants were less likely to be depressed at Time 2 and had a stronger sense of control. The Time 1 sense of control through perceived constraints predicted depression at Time 2, *w* = 0.45. Overall, these data show that public health restrictions and the sense of control are linked and that the sense of control has a powerful and long-lasting effect on depression status in restricted conditions, even once these have been lifted.

## 1. Introduction

### 1.1. Background

The COVID-19 pandemic has unsurprisingly seen increasing levels of depression [1]. Although designed to protect citizens and prevent illness, the side effects of public health restrictions of social isolation, withdrawal from normal working, and social, educational and leisure activities, e.g., [2], all might be predicted to contribute to psychological distress and increased incidence of depression. In addition, restrictions were often lifted and reimposed as the waves of the virus struck, with both individual and organisational plans often changing unexpectedly. Moreover, the reliability of information and advice given to the general public was publicly questioned in many jurisdictions. All of this would be predicted to create a sense of uncertainty that could have a profound emotional impact [3], with evidence suggesting that a strong sense of personal control protected against depression and psychological distress during the pandemic [4,5,6]. Here we hypothesise that a reduced sense of control is one of the pathways through which levels of depression increased during the pandemic due to the experience of public health restrictions imposed during COVID-19 previously being demonstrated to decrease people’s sense of control [7].

The aim of this study is to explore the relationship between the experience of public health restrictions, sense of control and depression. This is important in terms of increasing our understanding of environmental correlates of depression but also from the perspective of constructing public health restriction regimes to be as supportive as possible to mental health. First, we provide a brief overview of relevant studies.

The first year of the pandemic saw a number of studies published that documented the *immediate* impact of the pandemic on mental health. Higher levels of depression and anxiety were reported in data that were collected in the first months of the pandemic [8,9,10,11,12] and as early as two weeks after first cases reported in that jurisdiction [13]. A meta-analysis of worldwide studies published between 1 January and 8 May 2020 showed the prevalence of depression to be seven times higher than would typically be the case [1]. Furthermore, longitudinal data from the European COVID Survey, from two time points at the midpoint of the pandemic (November 2020 and April 2021), indicated that the prevalence of ‘probable depression’ was at 26% at both time points and particularly high in adults who were under 30 years of age. Overall, studies show a consistent picture of increased levels of depression that was evident from early on in the pandemic [14] and had not dissipated after 16 months (as measured from 1 January 2020) [1], although see [14].

The psychological impact of this pandemic has been studied from the perspective of the impact of quarantines and restrictions during previous public health emergencies (including Ebola, SARS, influenza, MERS, etc.). For example, Brooks et al. [15] conducted a review of relevant papers and described negative psychological impacts, including boredom, confusion, anger, concerns about supplies and finances, frustration and fear. Similarly, Tolares et al. [16] reviewed COVID and pre-COVID studies to examine the psychological impact of the situation, with a focus on those who have been infected, families, healthcare workers and the wider community. They argue that the focus on infection may reduce attention to the significant psychosocial impacts, including isolation, fear and vulnerability. Importantly, they make recommendations on the provision of public health information but also describe how a long-lasting health impact could result and recommend the need for supportive psychological interventions.

It would be important for any psychological intervention to target the processes through which negative psychological effects occur. Our focus in this study is on depression and evidence that a key correlate of depression, the sense of control, moderated the depressive experiences during the pandemic. For example, a number of cross-sectional studies have shown that a strong sense of control was protective in relation to depression during the pandemic [4,5,6]. In addition, Senan et al. [7] found that, for participants based in Saudi Arabia, the experience of restrictions, both their impact on daily life and their magnitude (i.e., the number that participants found distressing) positively predicted higher levels of depression. However, the sense of control moderated this effect, with a strong sense of control reducing the impact of restrictions on depression. Several other studies have also demonstrated a strong relationship between sense of control and depression, including personal control over health and treatment [17] and being an actor or director of one’s own life [18], as well as both internal mastery and external constraints [6]. These findings support a link between the subjective experience of restrictions, the sense of control and depression. However, none of these studies examined control and depression both during and after lockdown. This study, therefore, is important in examining longer-lasting effects of a pandemic, both on sense of control as well as depression. This paper has further followed the STROBE guidelines for cross-sectional/observational research and includes all relevant information [19].

### 1.2. Current Studies

The majority of the studies, described above, report data that were collected cross-sectionally in the early months of the pandemic (January to December 2020). There is limited evidence regarding the lasting psychological effects of the pandemic although see, [4,12,14,20] and the impact of public health restrictions in particular. It is possible that people become habituated to pandemic conditions and restrictions over time with less psychological distress ensuing over the longer term. In addition, longitudinal studies are required to explore whether the predictive relationships between restrictions, sense of control and depression ‘during’ a crisis time period are predictive of distress after the crisis has alleviated. This was the aim of the current study.

For these reasons, we collected data at two time points. The first time point was at a time in which restrictions had been in place for approximately 2 years and there was great uncertainty about the future. The second data collection took place when all restrictions had finally been lifted.

Across the two data collections, there were three overarching hypotheses:

**Hypothesis** **1 (H1).**
*The subjective experience of restrictions will be associated with the sense of control and depression status;*


**Hypothesis** **2 (H2).**
*The subjective experience of restrictions, the sense of control and depression status will change over time;*


**Hypothesis** **3 (H3).**
*The subjective experience of restrictions and the sense of control at Time 1 will predict depression status at Time 2.*


## 2. Materials and Methods

### 2.1. Ethical Statement

Ethical approval was received from the Ethics Committee of the Faculty of Education and Health Sciences at the University of Limerick [2021_12_25_EHS (ER)].

### 2.2. Context

The first survey data collection was undertaken during a period of great uncertainty in Ireland (Time 1, 7 January to 22 February 2022). Various levels of COVID-19 restrictions had been in place since March 2020 (c. 2 years), and there had been prior optimism in the autumn of 2021 that all restrictions would be lifted by 28 October 2021. However, on “…3 December, the Government reintroduced restrictions, including all nightclubs to close until 9 January 2022. Further restrictions were announced on 17 December 2021, including an 8 p.m. closing time for bars, restaurants, live events, cinemas and theatres until 30 January 2022.” On 22 January 2022, the Government removed most COVID-19 restrictions. Face masks and protective measures in schools remained in place until 28 February 2022.” At the time of Time 2 survey data collection on 17 May 2022 and since 1 April 2022, there were no public health restrictions in place in Ireland. This second data collection was designed to check whether there were changes in the study variables between Time 1 and Time 2 and whether there were lasting effects of the trends identified at Time 1.

### 2.3. Participants

Volunteers were adults aged 18 and over recruited to participate in an online survey via an invitation sent to email networks at a university in the Republic of Ireland and associated social media sites.

All volunteers who completed the survey at Time 1 were asked to indicate whether they would consent to be contacted within 6 months to repeat the survey. Those who gave consent were subsequently contacted by email to invite participation in Time 2 data collection.

#### 2.3.1. Time 1 Participants

A total of 474 volunteers opened the Time 1 survey, and 314 (66.24%) completed all survey items. Of these, 113 (36%) described themselves as male, 192 (61%) as female, 4 as non-binary, and 4 preferred not to say or self-describe. Participant ages ranged from 18 to 76 (M = 27.79, SD = 11.167), with 67% (*n* = 211) reporting living or staying with up to 4 people at least half of the time. Of the 314 participants who completed the survey in its entirety, 172 (54.78%) indicated that they would be prepared to be contacted within 6 months in order to repeat the survey.

#### 2.3.2. Time 1 and 2 Participants

Time 1 participants, who gave consent to be contacted, were emailed on 17 May 2022 and invited to complete the Time 2 survey. Of the 172 who agreed to be contacted again, 69 (40.12%) participants responded to the Time 2 invitation and began to complete the survey. A total of 47 (27.33%) participants completed all items, reached the end of the survey and provided information such that their data could be matched to the Time 1 data. Of these, 14 (30%) participants described themselves as male, 31 (66%) as female, 1 as non-binary, and 1 participant preferred to self-describe. Participant ages ranged from 19 to 51 (M = 29.57, SD = 10.314), with 53% (*n* = 25) reporting living or staying with up to 4 people at least half of the time. There was an average of 123 days (M = 123.66, SD = 5.76) between the Time 1 and Time 2 completion. See Appendix A for a flowchart of participant recruitment.

### 2.4. Measures

Demographic data were collected, including age (years), gender and number of members of the household.

#### 2.4.1. Beck Depression Inventory BDI: [21]

The original version of the BDI is a commonly used and well-validated and reliable self-report measure of depression in a wide range of populations [22] and in other similar studies [7]. The scale comprises 21 items related to depression symptoms and these are scored on a scale from ranging 0 (normal) to 3 (extreme mood), giving a total score from 0 to 63, where higher scores on the questionnaire indicate high levels of depressed mood. At Time 2, there were a small number of missing data points for three participants (*n* = 1, 2 items; *n* = 2, 1 item). Using Gale and Hawley’s [23] weighted mean method, these data points were replaced. Cronbach’s alpha showed that the scale has high reliability (T1 *α* = 0.91; T2 *α* = 0.93).

The primary outcome measure used in this study is the binary variable, depression status, categorised using BDI score standard cut-offs: scores of 10 were categorised as normal mood, low BDI; and scores of 11 or above were categorised as indicating some evidence of depressive symptoms, high BDI.

#### 2.4.2. Sense of Control Scale SOC: [24]

The Sense of Control Scale is a well validated self-report measure of personal control with good internal consistency [24] and has been used in several studies, e.g., [25,26]. The scale consists of 12 Likert items rated from 1 (strongly disagree) to 7 (strongly agree). Eight of the items relate to external factors that influence personal control, labelled perceived constraints, and four items relate to internal factors that influence personal control, labelled perceived mastery. Scores for the constraint subscale were produced by reverse scoring the perceived constraints items. For both subscales, the mean of the individual items is taken such that higher values represent a stronger sense of control. Reliability was excellent. For Time 1, Cronbach’s a *α* gave 0.775 and 0.859 for mastery and constraints, respectively, with similar values at Time 2 (Mastery *α* = 0.867, constraints *α* = 0.906) [26].

#### 2.4.3. Experience of Public Health Restrictions PHR: [7]

This measure was developed to explore the experience of public health restrictions as part of a study conducted in Saudi Arabia and was linked to public health restrictions in place in that country at the time. For the purposes of this study, we adapted the items to be consistent with restrictions in place in the Republic of Ireland.

This measure has two components:

##### Restriction Impact (0–50)

This was the sum of participants’ ratings of the extent to which public health measures affected their (1) daily life, (2) social life, (3) work, (4) daily choices and (5) weekly choices. Each item was rated from 0 (not at all) to 10 (completely), identical to Senan et al. [7]. Cronbach’s alpha values showed that this measure had good reliability (Time 1 *α* = 0.846; Time 2 *α* = 0.779).

##### Restriction Number

This measure was adapted from the Saudi study to match with public health restrictions in place in Ireland. Participants were presented with a list of 16 public health restrictions, and indicated which, if any, made them feel anxious, depressed or sad. These were: difficulties visiting family, difficulty socialising with friends, wearing a mask, sanitising hands frequently, unable to travel or difficulty travelling, social distancing measures, financial implications, job changes including remote working, unable to study, distance learning, job loss, curfews, requirement for antigen testing, requirement for PCR testing, requirements to isolate and requirements to quarantine. Participants were also given the opportunity to indicate if there were any other restrictions that made them feel depressed, anxious or sad and describe these. The count of these items was the participants’ score on this variable. With a maximum of 3 additional items, the maximum score for this measure was 19.

##### Time 1 versus Time 2 Adaptation

At Time 1, all items were worded in the present tense. However, at Time 2 and as restrictions had been lifted, the same items were rewritten in the past tense to provide a retrospective rating, i.e., “…Please rate how much public health restrictions affect**ed** your life…” and “…Please indicate which, if any, of the public health restrictions **made** you feel anxious, depressed or sad…” [emphasis in bold added].

### 2.5. Procedure

Participants were recruited through various social media sites such as Twitter and Facebook as well as university-wide emails between 7 January and 22 February 2022. All volunteers accessed the surveys by clicking on a link to participate in the survey that was hosted on Qualtrics. The first page of the survey was an information sheet that provided details about the study as well as an assurance that participants were allowed to cease participation at any time by closing the browser and their data would not be used in the study. The consent form required participants to clarify that they were over the age of 18, had read the information sheet and understood that they could quit the survey at any time. Then, demographic information was gathered—aspects such as gender, age, etc.—before participants proceeded to the questionnaires measuring the outcome and predictor variables. If participants agreed to be contacted again, they were asked to provide their preferred email address for the future contact. Those who agreed to be contacted again were sent a link to the same study with the same questionnaires on 17 May 2022 also hosted by Qualtrics. The same procedure was repeated for Time 2 as for Time 1 in terms of consent and questionnaires. This allowed us to not only examine relationships between variables at each time point, but also determine how scores changed over time. The change over time is of particular importance as Time 1 data were collected during national lockdown, whereas the restrictions had been lifted by Time 2.

### 2.6. Analytic Strategy

Preliminary measures t-tests were computed to compare the low and high BDI status group on the 4 predictor variables. Figures were constructed showing the prevalence of depression for different values of the predictor variables.

Hypothesis 1 (relation of perceived restriction variables and sense of control to BDI status) was evaluated in two stages for Time 1. In stage 1, a simple sequential binary logistic regression with forward selection (Wald) was conducted with BDI status (low, high) as the outcome variable, and sense of control mastery, sense of control perceived constraints, public health restrictions impact and public health restrictions number, as continuous predictor variables. The joint effects of all main effects and the following hypothesised 2-way interactions were evaluated (Mastery **×** Impact, Mastery **×** Number, Constraints **×** Impact, Constraints **×** Number). In stage 2, Generalized Linear Models were conducted with BDI status as the outcome and ONLY the predictors that were significant in the sequential logistic regression as predictors. Both binary logistic and Poisson lognormal analyses were conducted with model and with robust variance. The model with the best goodness of fit was retained.

H2 (change of measures over time) was evaluated by analysis of variance (ANOVA) with time as a repeated measure for each of 5 response variables (BDI total score, constraints, mastery, impact and number of restrictions. Power to detect a medium effect size was 84% (where α was set to 0.01 as there were 5 hypotheses).

H3 (effect of Time 1 predictor variables on BDI status at Time 2) was evaluated with a similar strategy to that used for H1. The first stage had exactly the same predictors as for H1, but the outcome variable was BDI status at Time 2. Subsequent generalized linear models enabled comparison of Poisson and logistic distributions. The power is weak with only 47 participants, so negative effects are food for further investigation, and should not be considered robust evidence.

## 3. Results

### 3.1. Time 1 Results

#### 3.1.1. Descriptive Measures

There were a total of 314 participants included in the Time 1 analyses. The average BDI score of the Time 1 sample was 15.28 (SD = 10.30, SE = 0.58), with *n* = 195 scoring 11 or above in the mild-to-moderately depressed category. The impact of restrictions was reported as moderately high (M = 32.17, SE = 0.57, SD = 10.11), with around half of the specified restrictions reported as making participants feel anxious, depressed or sad (M = 8.84, SE = 0.22, SD = 3.96). Sixty participants added items that were not included on the list but were distressful. These included vaccination and its implication, COVID-19 passports, the right to make choices, catching COVID-19 (feelings of shame), the inability to carry out activities that were an important part of their identity, such as sport and performing, the ability to develop new intimate relationships and make new friends. Despite all these signs of low levels of wellbeing from the impact of restrictions, the average sense of control was above the midpoint on both subscales (Mastery: M = 4.98 SE = 0.06, SD = 1.10; Constraints: M = 4.28, SE = 0.07, SD = 1.1 s).

#### 3.1.2. BDI Category, the Sense of Control and Perception of Restrictions

Table 1 shows descriptive statistics for all the predictor variables as a function of low and high BDI group. Participants in the low BDI category produced significantly higher scores on the sense of control subscales, and significantly lower scores on the measures of perception of public health restrictions (impact and number).

#### 3.1.3. H1. Joint Effects of Predictors on Probability of Being in High BDI Group

Table 2 shows the results for all variables with a significant predictive effect on BDI group (with results being the same using forward or backward variable selection). This shows that a higher number of restrictions, endorsed as distressful, and a lower sense of control (as measured by high perceived constraints) predicted a higher probability of scoring in the high BDI category.

In order to interpret the significant interactions, predictor variables were split into five categories and plotted against the predicted probabilities of being categorised at a member of the High BDI group. The interactions are illustrated by the different impact of restrictions for people with different levels of sense of control (see Figure 1).

Restrictions impact had the strongest effect on BDI, for people with average to high sense of control. Figure 1a showed that the impact of restrictions was strongest for average mastery ES simple effect = 0.07 and significant for all levels of mastery (all *p* < 0.001 to 0.029) except very low mastery (0.056). Figure 1b again showed that the strongest effects of restrictions were for those who have average, high or very high levels of control in relation to constraints (ES simple effects = 0.175 to 0.12) The probability of being categorised as depressed was highest for those with a low sense of control.

These findings are consistent with the hypothesis that those with a strong sense of control are much less likely to be categorised as high BDI than those with a weak sense of control. The predictive effect of perceived constraints (ES main effect = 0.425, *ES* Interaction = 0.194) was stronger than the predictive effect of mastery (ES Interaction = 0.161). Participants whose sense of control was weak in terms of high perceived constraints were equally likely to be in the high BDI group irrespective of the impact of restrictions. Those with a strong sense of control (low perceived constraints) and low impact of restrictions were the least likely to show evidence of depression. These findings suggest that the depressive impact of restrictions is more influential in those with a strong sense of control.

### 3.2. Time 2 Results

#### 3.2.1. Descriptive Measures

Of the 172 participants who agreed to be contacted again, 47 completed the survey at Time 2. The average BDI score in this sample was 9.87 (SD = 8.56, SE = 1.25), with *n* = 17 (36%) scoring 11 or above in the mild-to-moderately depressed category. The impact of restrictions reported retrospectively was moderately high (M = 31.79, SD = 8.84, SE = 1.29), with around half of the specified restrictions reported as having made participants feel anxious, depressed or sad (M = 7.49, SD = 3.85, SE = 0.56). Twenty-one participants (45%) added items that were not included on the list but were noted distressful. The average sense of control was above the midpoint on both subscales, Mastery: M = 5.34, SE = 0.16, SD = 1.09; Constraints: M = 4.94, SE = 0.18, SD = 1.26. High BDI participants produced significantly lower sense-of-control scores, across both the mastery and constraints subscales than low BDI participants (see Table 3), but there were no significant differences at Time 2 in the restrictions impact and number scores.

#### 3.2.2. H2. Time 1 versus Time 2

By using a repeated measures analysis of variance (ANOVA), we were able to compare scores on the BDI and predictor variables between Time 1 and Time 2 (see Table 4). Scores were indicative of a healthier state at Time 2. BDI scores were significantly lower at Time point 2. There were 7 participants that moved from the high to low BDI category, and only 2 moved from the low to high category, Cochran–Mantel–Haenszel *X*^2^ = 14.8, *p* < 0.001. Mean BDI Time 1 = 12.6 [9.9, 15.4], mean BDI Time 2 = 9.9 [7.3,12.5], *F*(1,45) = 9.3, *p* = 0.004. Sense of control as measured by perceived constraints was significantly higher at Time 2 in comparison to Time 1. Furthermore, the number of restrictions retrospectively endorsed as distressful was significantly less at Time 2 than Time 1.

However, mastery scores showed no significant difference between Time 1 and Time 2. Similarly, the impact of restrictions viewed retrospectively was not significantly different at the two time points.

#### 3.2.3. H3. Effect of Time 1 Measures on Probability of High BDI at Time 2

This effect was tested by entering Time 1 predictors (mastery, constraints, restriction impact, restriction number and the same hypothesised 2-way interactions as tested at Time 1) into a binary logistic regression model with Time 2 BDI category as the outcome variable. With both forward and backward regression to select significant effects, the only reliable predictor of high BDI group at Time 2 was Time 1 constraints, *B* = −1.17, Wald *X*^2^ (*df* = 1, *n* = 46) = 9.31, *p* = 0.002, ES_w_ = 0.45. Those with a low sense of control (high perceived constraints) at Time 1 were significantly more likely to be categorised as high BDI at Time 2.

Perceived constraints was the only Time 1 variable to predict BDI category at Time 2, such that for those with a lower sense of control in terms of the external forces that control their lives, perceived constraints were more likely to score in the high BDI category. Examples include having little control over things that happen to the individual or being unable to solve personal problems.

## 4. Discussion

This study was carried out during the first 5 months of 2022, in the Republic of Ireland, during which time public health restrictions were reintroduced (January 2022, Time 1) and subsequently lifted (April 2022, Time 2, May 2022). The results of the first data collection indicated that the self-reported *impact* of public health restrictions, in terms of daily and weekly activities and choices, was predictive of depression status. In the context of very challenging circumstances, the sense of control appeared to exert a protective effect, particularly in relation to the perception of perceived external constraints. However, those with the strongest sense of control appeared to be most vulnerable to the impact of public health restrictions. Importantly, whilst the internal sense of control, as measured by mastery, appeared to be static across the time points, the perceived constraints variable improved significantly once public health restrictions were lifted. With that being said, the sense of control in terms of the perception of constraints during the restrictions was a strong predictor of depression status once the restrictions were lifted. These findings will now be discussed in the context of our understanding of the impact of the pandemic on depression.

Since early 2020, studies have been published that have examined depression and other aspects of mental health during the pandemic. The consensus from these studies has been that depression rates have increased since the beginning of the pandemic [1]. Different correlates of depression status during the pandemic have also been examined, including gender [27], age [28], employment [29], risk of infection [9] and so forth, in order to map those at most risk under the circumstances. Others have examined putative protective factors, such as the sense of control [5], exercise [30] and affluence [11], with exacerbating factors including time spent watching the news [10] and exposure to personal stressors, such as job loss. Our findings add to this important body of work by elucidating at least one part of the psychological pathway from the pandemic environment of public health restrictions to depression. These data indicate that the sense of control during the crisis, in the form of perceived constraints in particular, is powerfully predictive of depression even when the crisis has passed.

These findings are consistent with decades of data that show that the sense of control has a lasting predictive value for depression [31] and provides a mediating pathway between stressful events and depression and anxiety [32]. Additionally, studies conducted during the pandemic have further demonstrated that having a low sense of control is related to higher levels of psychiatric symptoms such as depression and anxiety, e.g., [33]. This has been shown to hold true for many populations, including older adults [34], students [6,35,36], healthcare professionals [37], teachers, [38] and even yoga practitioners [17]. Various types of control have been measured during the pandemic, including the mastery and constraints variables we used here [26] as well as more specific measures that were more pandemic-related, such as personal control over health and treatment [39]. Similar to our study, Hamm et al. [40] demonstrated that depression declined across two months from April 2020 to June 2020, with those who shelved their goals being less vulnerable to depression and anxiety. Interestingly, disengagement with goals might be a means through which individuals ‘take control’ in uncontrollable circumstances and experience less psychological detriment.

Importantly, in this study, the sense of control, in terms of perceived constraints, as measured during the imposition of restrictions, predicted the likelihood of depression some 5 months later, when the crisis had passed. This was particularly interesting because of the interaction between the experience of restrictions and sense of control at Time 1. Participants with the weakest sense of control were highly likely to report depressive symptoms, so much so that there was no visible effect of restrictions. However, it was just those participants, who reported the strongest sense of control, for whom the experience of restrictions had a visible and deleterious effect on depression. This finding provides evidence of a link between the external environment (restrictions in this case), the sense of control (linked to external constraints) and then later depression. Furthermore, it suggests paradoxically that those who are at lower risk of depression due to their stronger sense of control are more vulnerable than we might have anticipated to the experience of restrictions, which may have compromised their strong sense of control.

### 4.1. Limitations and Strengths

In terms of limitations, it is not possible for us to say emphatically that restrictions affected the sense of control and thus depression. It is therefore possible that the sense of control influenced the experience of restrictions in the first place, or that depression affected all of these [41], with reciprocal causal pathways at play. In addition, the sample size for Time 2 data collection was relatively small; consequently, power was low and absence of significant effects is reason for further investigation. This study is, to our knowledge, the first to examine both during- and post-lockdown levels of depression and sense of control, and the findings are important. Additionally, the consistency between patterns of data, related to sense of control and depression, reported across studies is compelling. This study indicates the scope for further research.

Obviously, the sample has all the disadvantages of being an ‘opportunity’ sample and was inevitably limited to people who chose to respond. In particular, people with depression may well be overrepresented. Findings are of interest even if they only apply to a subset of people. It is also striking that these results were similar to those obtained in a culture and jurisdiction that implemented restrictions differently.

In terms of strengths, data were collected at two time points—when the restrictions had been in place for 2 years and when they were finally lifted—providing us with valuable, longitudinal insight. This is evidence for lasting mental health effects of the COVID pandemic. Further, no studies have been conducted in Ireland regarding sense of control and depression, and our study therefore can provide valuable information for this national population. We had no measurement or selection bias, nor was there any misclassification of data or results. As Ireland had similar lockdown protocols as other Western countries (REF), our study therefore has external validity in the Western world.

### 4.2. Implications for Mental Health

These findings have implications for mental health and offer opportunities for intervention and recommendation. For example, they suggest that if restrictions are structured and communicated such that people perceive fewer and lower impacts of ‘fixed’ constraints, then the restrictions should be less disadvantageous to mental health. It could also be argued that public health messaging could be adjusted to engender more empowering personal messages that do not compromise the sense of control so strongly. Somewhat counterintuitively, public health restrictions affected those with a high sense of control most strongly, making these participants vulnerable to depression.

## 5. Conclusions

These findings suggest that the sense of control could be a focus for mental health support and intervention in contexts, such as this pandemic, in which loss of control is a strong feature [42]. Overall, the findings are consistent with many years of research indicating that a strong sense of control is healthy and that loss of control can be harmful. They add to existing knowledge by including a sophisticated exploration of the experience of restrictions and assessing the longevity of their effects. Further, these findings suggest possible meaningful interventions.

## Figures and Tables

**Figure 1 ijerph-19-14429-f001:**
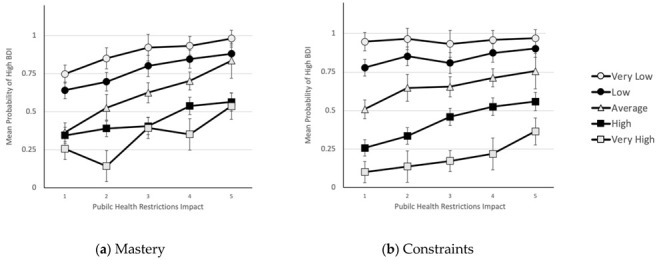
Mean predicted probability of high BDI group as a function of the impact of public health restrictions and sense of control where panel (**a**) shows the interaction with Sense of Control: Mastery; and panel (**b**) shows the interaction with Sense of Control: Constraints. Error bars correspond to the standard errors of the mean.

**Table 1 ijerph-19-14429-t001:** Summary of predictor variable means for low and high BDI participants.

BDI Category	Low BDI				High BDI				Independent *t*-Test
Predictor Variables	M	SE	95%	95%	M	SE	95%	95%	
			LCL	UCL			LCL	UCL	
Mastery	5.53	0.08	5.38	5.68	4.64	0.08	4.48	4.79	7.61 *
Constraints	5.15	0.08	4.99	5.31	3.76	0.07	3.61	3.9	12.48 *
Restriction Impact	29.03	0.89	27.26	30.8	34.08	0.71	32.69	35.48	4.42 *
Restrictions Number	7.27	0.38	6.52	8.02	9.8	0.25	9.3	10.3	5.77 *

Data are mean values (M) and standard errors (SE) with lower and upper 95% confidence limits (LCL, UCL); *df* = 312, * all *p* < 0.001.

**Table 2 ijerph-19-14429-t002:** Binary logistic regression final model for BDI (low, high).

Variables	B	SE	Wald	*df*	*p*	Exp(*B*)	ES_w_
1. Restrictions number	0.104	0.043	5.79	1	0.016	1.11	0.136
2. Constraints	−2.117	0.281	56.79	1	<0.001	0.12	0.425
3. Mastery × Restrictions Impact	−0.017	0.006	8.1	1	0.004	−0.017	0.161
4. Constraints × Restrictions Impact	0.028	0.008	11.83	1	<0.001	0.028	0.194
Constant	7.965	1.153	47.73	1	<0.001	2877.17	

NB. The interaction between variables is indicated by the I symbol. The effect size measure w is referred to as ES_w_ and was calculated as √ (Wald/N).

**Table 3 ijerph-19-14429-t003:** Summary of predictor variable means for low and high BDI participants.

BDI	Low BDI				High BDI					
Predictor	M	SE	LCL	UCL	M	SE	LCL	UCL	F	*p*
Mastery	5.70	0.17	5.35	6.05	4.69	0.26	4.14	5.24	11.39	0.002
Constraint	5.56	0.16	5.22	5.89	3.85	0.26	3.3	4.39	34.72	<0.001
Impact	31.93	1.26	29.37	34.50	31.53	2.86	25.47	37.59	0.02	0.882
Number	7.23	0.77	5.67	8.80	7.94	0.78	6.29	9.59	0.36	0.551

Data are mean values (M) and standard errors (SE) with lower and upper 95% confidence limits (LCL, UCL); F derived from multivariate ANOVA, with BDI as the Independent Variable, with *df* = 1, 45.

**Table 4 ijerph-19-14429-t004:** Comparisons between T1 and T2.

Predictor	Time	M	SE	LCL	UCL	F	*p*	Cohen’s d
BDI	1	12.51	1.34	9.81	15.21	9.12	0.004	0.45
	2	9.88	1.25	7.36	12.39			
Mastery	1	5.20	0.16	4.88	5.52	1.70	0.199	−0.19
	2	5.34	0.16	5.02	5.66			
Constraints	1	4.64	0.18	4.29	5.00	7.79	0.008	−0.41
	2	4.94	0.18	4.57	5.31			
Impact	1	32.68	1.38	29.9	35.47	0.49	0.486	0.10
	2	31.79	1.29	29.19	34.38			
Number	1	8.49	0.63	7.22	9.75	5.78	0.020	0.33
	2	7.50	0.56	6.36	8.62			

Data are mean values (M) and standard errors (SE) with lower and upper 95% confidence limits (LCL, UCL); *df* = 1, 46.

## Data Availability

The data will be available for readers to download at the following DOI. Reserved DOI: doi:10.17632/5542prtpj4.1.

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
