# Peer review of "Sense of Control and Depression during Public Health Restrictions and the COVID-19 Pandemic"

_ijerph, 2022, doi:10.3390/ijerph192114429_

Round 1
Reviewer 1 Report
It's a meaningful research and may hold the reader's interest. Some details of writing need attention, such as the format of the references.
Moreover, the sample size was too small. At T2, only 47 people took part in the study.
Author Response
Reviewer 1.
It's a meaningful research and may hold the reader's interest. Some details of writing need attention, such as the format of the references.
Moreover, the sample size was too small. At T2, only 47 people took part in the study.
We have carefully reviewed and amended the writing in order to produce a clearer account of the study, and to carefully consider the implications, and indeed the limitations, including small sample size. We also include a power analysis in the new section on Analytic strategy.
Reviewer 2 Report
Msetfi et al. aimed to examine the relationship between the experience of public health restrictions, the sense of control 12 and depression, both during and after restrictions were lifted
It is an interesting article but there are some shortcompings that need attention. I would like to make some suggestions and contributions to the present manuscript. The authors need to check that their manuscript complies with STROBE guidelines for observational research: https://www.equator-network.org/reporting-guidelines/strobe/ A statement needs to be added to the manuscript confirming the same.
Methods
-The authors should explicitly mention the study design. In addition, no information is provided on sample estimation and sampling techniques used.
-Add the selection criteria used to recruit participants in the two measurements.
-Add a flowchart of participant selection at the two moments of the research.
-In section 2.3.1, the authors should mention the percentage of participants who completed the survey (314/474).
-In section 2.3.2, the authors should mention the percentage of participants who completed the survey (69/172).
-In instruments, the authors should add values for internal consistency and prior antecedent validity.
-The procedures have been described in a very limited way; more detail is required on the processes carried out during the research.
-The statistical tests used for the analysis of the research have not been described. This information is extremely important in order to be able to evaluate whether the results section has been carried out correctly.
Results
-I suggest that you improve the presentation of your results. You could start by describing the general variables of the population, instead of mentioning it in the methods section (a section that has serious limitations).
-You should specify whether the logistic regression model presented in Table 2 corresponds to the simple or multiple model. In addition, it is suggested that this analysis be rethought, according to the correct measure of association that the authors should report. The odds ratio should be estimated mostly in case-control studies. In this analysis, prevalence ratio (PR) estimation would be more appropriate. It is suggested to use generalized linear models (GLM), Poisson distribution family, robust variance, and to report PR and CI of the single and multiple models.
-It is not understood whether Table 6 within the results section is really useful. Please, review.
Discussion
-I suggest that they improve the presentation of their discussion. They should expand their discussion with findings from similar studies, to mention whether their findings are really similar/opposite to that of this study. Additionally, they should explain the significant differences found in the independent variables evaluated.
-Authors should add a "Relevance of findings to mental health" section. Additionally, they should reinforce their limitations paragraph and explicitly mention if their research presents measurement bias, selection, misclassification, among others; they should also mention how they have solved these limitations and if these problems invalidate the validity of their research. Finally, they should mention the main strengths of the study.
Author Response
Reviewer 2.
Suggestions for Authors
Msetfi et al. aimed to examine the relationship between the experience of public health restrictions, the sense of control 12 and depression, both during and after restrictions were lifted
It is an interesting article but there are some shortcomings that need attention. I would like to make some suggestions and contributions to the present manuscript. The authors need to check that their manuscript complies with STROBE guidelines for observational research: https://www.equator-network.org/reporting-guidelines/strobe/ A statement needs to be added to the manuscript confirming the same.
We have now confirmed that this manuscript complies with STROBE guidelines and have added a statement conforming this (line 90, p .2).
Methods
-The authors should explicitly mention the study design. In addition, no information is provided on sample estimation and sampling techniques used.
-Add the selection criteria used to recruit participants in the two measurements.
-Add a flowchart of participant selection at the two moments of the research.
Study design is now explicitly stated, with participant sampling information and criteria clearly described in the method section. A flowchart has also been included.
-In section 2.3.1, the authors should mention the percentage of participants who completed the survey (314/474).
-In section 2.3.2, the authors should mention the percentage of participants who completed the survey (69/172).
All percentages are now included.
-In instruments, the authors should add values for internal consistency and prior antecedent validity. The values for internal consistency are provided in the measures section, with fuller information provided where relevant.
-The procedures have been described in a very limited way; more detail is required on the processes carried out during the research. We have provided a much fuller, more detailed description of the processed involved in data collection and analysis.
-The statistical tests used for the analysis of the research have not been described. This information is extremely important in order to be able to evaluate whether the results section has been carried out correctly.
We have now included a section which describes the analytic strategy in much more detail.
Results
-I suggest that you improve the presentation of your results. You could start by describing the general variables of the population, instead of mentioning it in the methods section (a section that has serious limitations).
We have now included descriptive means Tables for both time 1 and time 2
-You should specify whether the logistic regression model presented in Table 2 corresponds to the simple or multiple model.
We have now clarified that this analysis is a simple regression as response variable has only 2 categories. There are multiple predictors and we report effects of those of the multiple predictors that are significant.
In addition, it is suggested that this analysis be rethought, according to the correct measure of association that the authors should report. The odds ratio should be estimated mostly in case-control studies. In this analysis, prevalence ratio (PR) estimation would be more appropriate.
The use of odds ratios in generalized linear models is not limited to case-control studies. It is prevalent as an effective measure for inferential tests wherever the response variable is a binary proportion.
However we agree that prevalence is most informative for descriptive measures and it is prevalence measures that are shown in the figures.
It is suggested to use generalized linear models (GLM), Poisson distribution family, robust variance, and to report PR and CI of the single and multiple models.
Thanks for this useful suggestion. We have now done that, but it turns out that logistic model is better.
It is now explained in the analytic strategy section that that strategy was in 2 stages. (1) identify significant predictors using stepwise facilities in SPSS. (2) Use generalized linear models only on significant effects. There are theoretical reasons, as reviewer perceptively notes, that the distribution might be Poisson. Hence, we also tested Poisson. However, goodness of fit was better. For H1 at time 1 logit Aikake = 277, Poisson Aikaike = 541. For H3 at time 2 logit Aikake = 36, Poisson Aikaike = 56. Obviously GLM and logistic regression give identical results if predictors are identical and distribution is logit. However, the sequential procedures in logistic regression are particularly useful. GLM has advantage of potential of robust variance, but for our data it made no difference
-It is not understood whether Table 6 within the results section is really useful. Please, review.
We agree that the table is superfluous. We have added a sentence instead that gives an example of what the perceived constraints are.
Discussion
-I suggest that they improve the presentation of their discussion. They should expand their discussion with findings from similar studies, to mention whether their findings are really similar/opposite to that of this study. Additionally, they should explain the significant differences found in the independent variables evaluated.
We have added to the discussion section and expanded our discussion on findings from similar studies in relation to the variable evaluated in the present study.
-Authors should add a "Relevance of findings to mental health" section. Additionally, they should reinforce their limitations paragraph and explicitly mention if their research presents measurement bias, selection, misclassification, among others; they should also mention how they have solved these limitations and if these problems invalidate the validity of their research. Finally, they should mention the main strengths of the study.
We have now included a section on relevance of findings to mental health, and have fully addressed biases and so forth which might be present.

Round 2
Reviewer 2 Report
The authors have corrected the observations and the article has been substantially improved.